# Implementing the EHRI-KG: From declarative mapping rules to a three-layered KG construction architecture

Herminio García-González[1,*], Mike Bryant[2]

[1]*Kazerne Dossin, Goswin de Stassartstraat 153, 2800 Mechelen, Belgium*

[2]*NIOD Institute for War, Holocaust and Genocide Studies, Herengracht 380, 1016 CJ Amsterdam, The Netherlands*

## Abstract

Conducting transnational research with primary source material relating to the Holocaust can be a complex endeavour, mainly due to the fragmentation and dispersal of much important material across different holding institutions and jurisdictions. The EHRI Portal acts as an aggregator of archival descriptions with the aim of lowering the barriers to the discovery of and access to source material. Building on earlier work, we set out to offer this data in a format more amenable to semantic technologies, as well as aligned with the ICA's new Records in Contexts standard for archival description. In this paper, we explore the EHRI Knowledge Graph three-layered architecture (batch conversion, validation and continuous update) and reflect on the challenges encountered and the solutions involved, to finally establish a discussion on further needs and future points of development within the Knowledge Graph Construction community and its toolset. We believe that this work can be helpful for other practitioners seeking to implement similar architectures, while serving as a use case for future work which will allow users to further streamline their developments and reduce the overall implementation time.

## Keywords

Knowledge Graph construction, architecture, declarative mapping rules, EHRI, ShExML

## 1. Introduction

Archival material and the primary sources it represents are a key part of the historical research process and its preservation, classification, and cataloguing is fundamental to ensuring access to our collective memory. In the field of Holocaust studies and Holocaust historiography, however, the intentional destruction of evidence and the massive dislocation of people during and after the Second World War introduced particular challenges for researchers, due to the fragmentation and scattering of many important sources throughout widely dispersed collection-holding institutions. To meet this challenge, the European Holocaust Research Infrastructure (EHRI) undertook the creation of the EHRI Portal[1], an aggregator of Holocaust-relevant archival descriptions which seeks to serve as an entrypoint for Holocaust researchers [1]. In a recent project, we explored the possibility of aligning the data in the EHRI Portal to a format more amenable to the Semantic Web standards [2], adopting the (then draft) version of the Records in Contexts Ontology (RiC-O) [3] and seeking to improve the general FAIRness of the data [4]. Since then, the International Council on Archives (ICA) Expert Group on Archival Description (EGAD)[2] has released the first stable version of RiC-O[3] and EHRI has launched the EHRI-KG project[4] with the aim of consolidating and improving the Linked Open Data (LOD) transformation infrastructure into a production-ready service.

In this paper we present the data mapping architecture and workflow alongside some interesting challenges and the solutions we have derived. While the process heavily relies on declarative mapping rules [5] (due to their generally favourable maintainability), many orchestration components and *ad-hoc*

*KGCW'26: 7th International Workshop on Knowledge Graph Construction, May 10–11th, 2026, Dubrovnik, Croatia*

*Corresponding author.

✉ herminio.garciagonzalez@kazernedossin.eu (H. García-González); m.bryant@niod.knaw.nl (M. Bryant)

🌐 https://herminiogarcia.com/ (H. García-González); https://www.niod.nl/en/staff/mikebryant (M. Bryant)

🆔 0000-0001-5590-4857 (H. García-González); 0000-0003-0765-7390 (M. Bryant)

[1]https://portal.ehri-project.eu/
[2]https://www.ica.org/ica-network/expert-groups/egad/
[3]https://github.com/ICA-EGAD/RiC-O/releases
[4]https://ehri-kg.ehri-project.eu/

solutions are still required. We believe that this architecture can inform other practitioners facing similar situations while, at the same time, prompting discussions around new required features in the Knowledge Graph Construction (KGC) community.

The rest of this paper is structured as follows: Section 2 describes the related work, in Section 3 we introduce the EHRI Portal and its technical stack, followed by a description of the conversion architecture in Section 4; Section 5 discusses the additions required to the declarative mapping technology as a motivation for the their adoption and implementation within the state-of-the-art solutions; finally, in Sections 6 and 7 respectively we outline our future plans and draw some conclusions from this work.

## 2. Related Work

Knowledge graphs are being increasingly adopted in fields such as cultural heritage and historical research [6], life sciences [7] or industrial applications [8], and their semantics, enhanced interoperability, and the capacity to represent complex relations in a graph structure have been demonstrated to be particularly effective in solving a variety of challenges.

The creation and maintenance of KGs requires the combination and harmonisation of various different techniques, leading to the generation of methodologies formalising some of these approaches, along with tools to tackle laborious aspects. Two prominent examples in the field are ontology and KG engineering. However, some studies have recently claimed that a more comprehensive and extensive methodology is necessary in order to ensure a more harmonised and homogeneous KG lifecycle [9, 10].

While many works describe the creation of KGs and their architectures, only a few provide sufficient detail for establishing a comparison with the current approach. In [11] the authors report on the creation of a KG for the discovery of interactions between drugs used for COVID-19 treatment, created using a combination of RML [12] mapping rules and Natural Processing Language (NLP) techniques. RML is also used by [13] to construct a KG for a public procurement data space whose execution is orchestrated by Apache Airflow[5]. Similarly, in [14] Apache Nifi[6] is utilised to manage the workflow for creating a commercial KG based on RML mappings. In order to offer a KG for the European Railway, in [15] YARRRML [16] rules (converted to their RML counterparts) were employed though no further details are given around the followed workflow. Also on the transport domain, the authors of [17] produced a multimodal KG using Chimera [18], a framework to implement data transformation pipelines covering the graph construction (lifting), transformation, validation and exploitation (lowering) phases. In [19] the construction of KGs for research-performing organisations is discussed for which YARRRML was employed through a workflow that ensured the creation of the ontology and the declarative mapping rules, the validity of the latter against the ontology and the database model, and the final KG materialisation. The authors elaborate further on some of the lessons learnt like the need for simple tools (like YARRRML with its compact syntax) or the need for separating the mapping rules by ontology classes (similar to the process that we describe in Section 4).

As a result, we can extrapolate that the creation of such KGs relies on some well-established tools and techniques (like RML and its myriad of conformant engines), yet some parts of the process are still developed in an *ad-hoc* manner. It is worth noting though that the authors of [10] have recently published the results of a survey[7] seeking to validate the proposed KG lifecycle methodology through the coverage of each of the defined activities by 31 ongoing projects dealing with the creation of KGs [20], helping to shed some light on commonalities and best practices.

In the Holocaust domain, several initiatives are seeking to provide access to Holocaust data as LOD through different thematic, regional, and institutional scopes. For example, Netwerk Oorlogsbronnen[8] covers WWII Dutch sources whereas Archiefpunt[9] enables access to archival descriptions in Flanders and Brussels but without any further topical filtering. Similarly, some institutions dedicated to the

---

[5]https://airflow.apache.org/
[6]https://nifi.apache.org/
[7]https://lot.linkeddata.es/LOT4KG/survey.html
[8]https://www.oorlogsbronnen.nl/
[9]https://archiefpunt.be/

Holocaust also offer their data – or part thereof – in Semantic Web formats, like CDEC [21] which offers a KG of Italian Holocaust Victims or the Resistance in Belgium[10] developed by CegeSoma/State Archives of Belgium on top of a Wikibase instance [22, 23]. Nevertheless, a broader implementation of semantic technologies in this field is still a work in progress with a very unbalanced distribution across regions and topics. As such, the EHRI-KG is, to the best of our knowledge, the first transnational effort to bring Holocaust archival descriptions together in a unified KG.

## 3. Preliminaries

The EHRI Portal started its development in 2012 with an architecture based on a Labelled Property Graph (LPG). Reasons for this included its ability to simplify the representation and querying of hierarchically-structured data, along with ease of evolving the data model over time [1]. The dearth of semantic ontologies well-suited to the archival problem space, as well as a lack of expertise in the consortium, also played a factor in this decision. Recent advancements in KGC technologies, however, and their use in the new ICA's archival standard, motivated us to explore the publication of EHRI's archival metadata as LOD.

From this standpoint, and taking into account that a wholesale reworking of the EHRI Portal's persistence layer would not be realisable due to time and money constraints, we searched for a method that would allow the project to maintain its usual activities with the current architecture, while providing a new method for accessing, querying, and linking its data. The solution was realised in the EHRI-KG[11], a live replica of the EHRI Portal based on semantic standards and fully queryable by third-party users via a SPARQL endpoint[12]. This solution not only solves our requirement of offering the data as LOD – and aligning with the new ICA's RiC-O standard – but also enables users to perform arbitrary queries without impacting our main platform.

While this solution may seem simple enough, the difficulties lay, as many times in this field, in the volume and variety of the data which, as described later, determined many of the technical decisions. The EHRI Portal holds, at the time of writing, more than 380,000 archival descriptions held by 2,318 institutions located in 61 countries which are enriched by a set of additional controlled vocabularies and links, all of them following existing archival standards (e.g, ISAD(G), ISAAR, ISDIAH, etc.). Despite it not being large enough to be considered Big Data [24], this amount of data, and in many cases its lack of normalisation, already imposed many organisational and technical challenges which are further described in Section 4 alongside the final architecture.

## 4. General architecture

At a high level, the architecture is divided into three main processes: a batch conversion workflow, a validation phase, and a module for handling update operations. Figure 4 provides a graphical representation of this three-layered architecture.

### 4.1. Batch conversion

The batch conversion layer is at the core of the conversion process. In this layer, the data is downloaded from the EHRI Portal, converted to RDF and merged into a single big RDF file which will be finally delivered to a triple store for stable persistence and serving other applications (through a SPARQL endpoint). This process is designed to be agnostic to the triple store technology, giving us the freedom to switch platforms depending on our specific requirements.

---

[10]https://data.arch.be/?lang=en
[11]https://lod.ehri-project-test.eu/
[12]https://lod.ehri-project-test.eu/query/

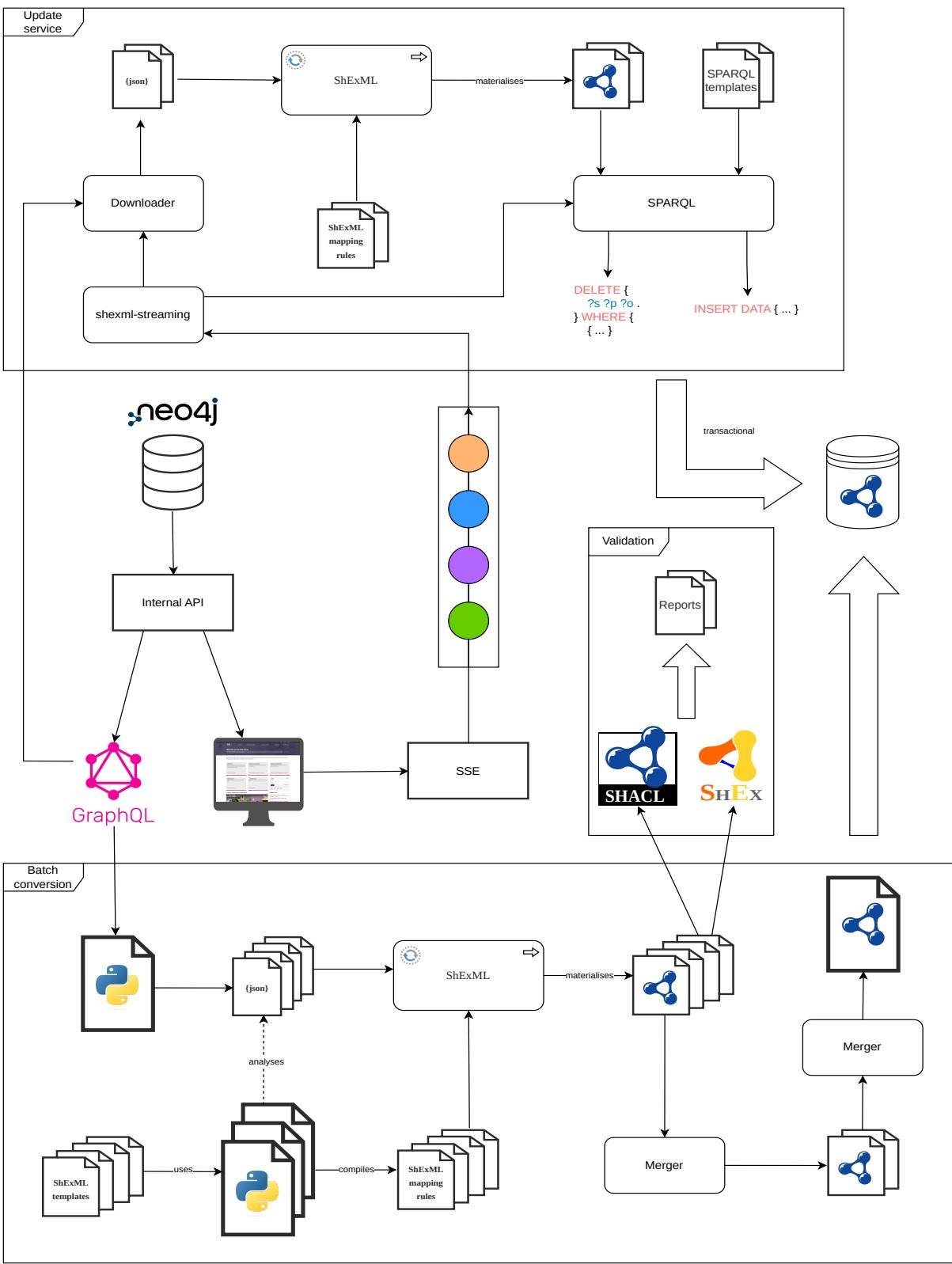

**Figure 1:** Diagram representing the implemented architecture. The central part represents the existing EHRI Portal stack, where a Neo4j graph database is accessed via an internal API which is leveraged by the EHRI Portal and the GraphQL endpoint. On the bottom, the batch system uses the GraphQL endpoint to perform conversion for targeted entities and materialises a comprehensive RDF graph which is later deposited in a triple store. On the top of the diagram, the update service connects to the SSE endpoint which transmits a series of events which are processed and trigger the different update transactions against the triple store. The validation layer uses the materialisation outputs before any merge. The arrows represent the sense in which the data flows within the architecture.

### 4.1.1. Downloader

This module downloads data for all the entities concerned in the conversion process, that is, countries, institutions, archival descriptions, people, corporate bodies and links (copy-original, creation and access points links). For each of these entities, a GraphQL query template is provided containing the attributes that we wish to convert. This template is then processed by the module which iterates over it for each result page delivered by the API. As a result, the process will create an output file in the JSON format for each page of an entity. Given that this process can take a bit of time, we introduced affordances for avoiding flooding the endpoint (in terms of idle time between downloads) and avoiding re-downloading the data that has already been fetched. The latter also allows for a resumption of the download process at any time and tolerates potential network issues.

### 4.1.2. Data mapping

In order to make this process as reusable, flexible, and understandable as possible we decided to use declarative mapping rules. As described later in this section, however, they cannot yet fulfil all requirements so additional orchestration code was needed.

We decided to use ShExML for the conversion based on a two-fold decision: 1) ease of use as described in [25] (in case non-experts needed to contribute) and 2) as one of the authors is the main developer of the ShExML engine, it allowed us to rapidly adapt the language and the engine with new features if required (some of the introduced additions will be explained later).

Thus, the conversion is based around a set of ShExML templates developed for each domain entity involved. This per-entity data mapping allows for a separation of concerns (not necessitating the re-conversion of data for other entities whenever a change is introduced) and for updating entities in an isolated fashion (as further explained in Section 4.3). Furthermore, the templates are modularised to increase reusability and readability, which required the addition of a new **IMPORT** directive[13] in ShExML. This directive is able to import other files including arbitrary ShExML contents to compose a single ShExML input prior to parse time.

Additionally, following the downloader's per-page breakdown, a set of mapping rules is created for each results page based on the previously created templates. This is delegated to a Python script which analyses the downloaded JSON files for an entity and creates the consequent mapping rules, embedding the correct source path in ShExML's **SOURCE** directive (see Listing 1 for an example).

Listing 1: Generated mapping rules for the first institutions' results page where the **SOURCE** is defined by the mapping rules creation script. The **IMPORT** clause allows for centralising the shared code (between the same entity's mapping rules) and performing modifications after their creation.

```
IMPORT <ShExMLTemplates/partial/RepositoriesHeader.shexml>
IMPORT <ShExMLTemplates/partial/MatcherLanguageCode2Digit.shexml>
SOURCE repositories <institutions/institutions_1.json>
IMPORT <ShExMLTemplates/partial/RepositoriesIteratorsAndShapes.shexml>
```

Right after the creation of all the mapping rules for an entity, the script calls the ShExML engine with the mapping rules as inputs and checks whether a specific page has already been materialised, as to avoid a re-materialisation and therefore support the resumption of the process at any point in time. As a performance affordance, the Python script allows running the mapping rules in parallel using half of the available CPU cores, for example, if the CPU has 8 cores – either physical or virtual – the script will divide the queued conversions in four chunks and launch four conversion threads simultaneously. The generated RDF files are then persisted in a dedicated folder for each entity.

---

[13]https://shexml.herminiogarcia.com/spec/#import

### 4.1.3. Merging results per entity

Once all pages for an entity are materialised we need to merge them in order to provide a single RDF file per entity. Based on the RDF compositional property one can just merge several RDF graphs without requiring any other major operation. However, in order to favour performance (and avoid re-serialising the data) we opted for a syntactic merge at the level of the file. As we materialise the data in the Turtle serialisation format (due to data size concerns) we need to remove and homogenise the prefix declarations to only preserve one set of declarations at the top of the merged file. This process is performed using the **`sed`** command line utility which processes the final file in a line-per-line streaming fashion, making the process very efficient. Listing 2 shows an example of this process for data relating to collection-holding institutions.

Listing 2: Excerpt of the Bash script used to merge all the institutions' Turtle files. `...` indicates additional content which has been omitted for brevity reasons.

```
cat shexmlOutputInstitutions/*.ttl > institutions.ttl
sed -i 's/@prefix.*//g' institutions.ttl

sed -i "1i\@prefix wd:    <http://www.wikidata.org/entity/> ." institutions.ttl
sed -i "1i\@prefix rdfs:  <http://www.w3.org/2000/01/rdf-schema#> ." institutions.ttl
sed -i "1i\@prefix ehri:  <http://lod.ehri-project-test.eu/ontology#> ." institutions.ttl
sed -i "1i\@prefix ehri_countries: <http://lod.ehri-project-test.eu/countries/> ." institutions.ttl
...
```

### 4.1.4. Merging all entities

Similar to the previous merge, this process combines the output files for different entities (generated by the previous module) into a single RDF Turtle file, again normalising prefix declarations. Additionally, it downloads the data for the EHRI vocabularies (i.e., controlled terms for subject headings, camps and ghettos), and given that they are already offered in RDF following the SKOS vocabulary, they are simply added into the merging process as an additional entity file. Finally, a set of links coming from entity linking activities is also incorporated at this stage, allowing us to keep this concern separated and making it independently updatable.

### 4.1.5. Further data mapping challenges

Even though this architecture addresses many of the issues arising from a lack of certain features in declarative mapping languages, and in ShExML specifically, there are still some challenges that are best addressed within ShExML in order to achieve a more sustainable solution. In this section, we highlight some of them and reason why they are more suited to be solved within the mapping language and/or engine, rather than external tooling.

**Matchers:** Albeit not being a new feature in ShExML, this functionality took a new dimension during this project. In essence, it allows to replace one pre-defined text occurrence for another, and chain these declarations into a single block for automatic processing in the generation of a particular object.[14] The main use case would be to match naming conventions between different KGs letting users link to other graphs in a seamless manner. However, in this specific project they were used to normalise language tags as the ones coming from the EHRI Portal follow the ISO 639-2[15] codes and we needed to align them with BCP 47[16], as prescribed by the RDF 1.1 specification[17], which favours the use of the shortest code from within the ISO 639 family. Listing 3 shows the language code matcher and Listing 4 exemplifies its use within the institutions' mapping rules.

---

[14]https://shexml.herminiogarcia.com/spec/#matcher
[15]https://www.loc.gov/standards/iso639-2/
[16]https://www.rfc-editor.org/info/bcp47
[17]https://www.w3.org/TR/rdf11-concepts/

Listing 3: Excerpt of the **MATCHER** used to convert ISO 639-2 language codes into BCP 47 conformat ones. . . . indicates additional content which has been omitted for brevity reasons.

```
MATCHER langcode_two_digit_equivalent <aar AS aa &
abk AS ab &
afr AS af &
aka AS ak &
alb AS sq &
...
>
```

Listing 4: Excerpt of the institutions' mapping rules which shows how the **MATCHER** in Listing 3 is leveraged to produce correct language tags. . . . indicates additional content which has been omitted for brevity reasons.

```
ehri:Institution ehri_institutions:[organization.id] {
  a ehri:Institution ;
  rico:name [organization.descriptions.name] @[organization.descriptions.languageCode MATCHING
      langcode_two_digit_equivalent] ;
  ...
}
```

**External functions:** The possibility to execute certain functions for transforming the output or conditionally generating a triple based on a certain condition is a ubiquitous functionality among different declarative mapping languages, we argue though that their implementation in the ShExML engine allows for a fast prototyping and extensibility. Unlike other languages, these functions are not gathered in a centralised hub/library nor defined in a declarative fashion (e.g., FnO [26]) but rather implemented in pure Scala code in an external file[18], similar to the mechanism proposed in [27] for RML using Python. This approach provides the full power of a Turing-complete language with a large standard library and allows for rapid prototyping due to the absence of intermediate abstraction layers and boilerplate. The ShExML engine will load these definitions and employ reflection to call the defined functions passing the relevant data as arguments of the function.[19] Listing 5 shows an example of a function declaration.

Listing 5: Example of the declaration of a function used to generate the IRIs for the different institutions' addresses.

```
class Transformers {
  def combinedPathInstitutionAddress(institutionID: String, index: String): String = {
    institutionID + "/addresses/" + (index.toInt + 1)
  }
}
```

**Built-in functions:** Whereas external functions can cover a lot of the use cases where an extension of ShExML is required, there are still some specific functionalities that are only easily solvable natively within the language and/or engine. During this project we identified one of those cases as we needed to recover the iteration index for creating a unique IRI for each generated entity in the iteration loop. For this purpose we initially tried to use the ShExML's autoincrement identifiers[20], along with the previously described external functions feature. Autoincrement IDs, however, were designed to work on a global scope and not capable of attending to nested contexts, whereas external functions were (by design) stateless and similarly not context-aware.

Relying on the query language for the iteration index, alternatively, can return ambiguous results as, for example, while XPath is able to retrieve information from the surrounding context (albeit in a complex manner) the same is not possible with JSONPath. Therefore, we came up with a new feature for the ShExML language called built-in functions[21], as they need to be aware of the current execution context and are, thereby, only implementable within the engine. For the moment only one of such

---

[18] https://shexml.herminiogarcia.com/spec/#functions
[19] https://shexml.herminiogarcia.com/spec/#invoking-functions
[20] https://shexml.herminiogarcia.com/spec/#autoincrement-ids
[21] https://shexml.herminiogarcia.com/spec/#built-in-functions

functions has been implemented, **index**(), but the mechanism was devised to be expandable (see Listing 6).

Listing 6: Excerpt of the institutions' mapping rules showing how the **index**() built-in function is used in combination with the function defined in Listing 5 to produce a unique IRI for each address. . . . indicates additional content which has been omitted for brevity reasons.

```
ehri:InstitutionAddress ehri_institutions:[transformers.combinedPathInstitutionAddress(organization.
    descriptions.addresses.repository_id, organization.descriptions.addresses.repository_id.index())]
    {
  a ehri:InstitutionAddress ;
  ...
}
```

## 4.2. Validation phase

In an effort to assess the validity of the outputs generated by the batch transformation, we developed a validation workflow which could highlight formatting and normalisation errors in the data. For this purpose, we employed a set of rules in SHACL and ShEx, drafted, in a first instance, automatically by the ShExML engine using its ability to generate RDF shapes from mapping rules and materialised data, and further fine-tuned according to our knowledge of the domain model. Consequently, a script was developed to validate each file (per entity and per results page) against the pair of shapes files and produced a consolidated report highlighting the number of errors and their type (see Listing 7). Following the same rationale used for the conversion process allows us to deliver a much more readable output than just validating against the entire graph and enables a fine-grained inspection of the input and output files relative to the mapping rules. This process has not only allowed us to improve some small details and to detect some human-induced errors in the mapping rules but rather more importantly to identify errors and non-normalised data in the EHRI Portal.

Listing 7: Excerpt of the validation report showing the validation status for some institutions' data and the total for this type of entity. This report is completed by another file that hosts the validation errors as outputted by the validation engine. . . . indicates additional content which has been omitted for brevity reasons.

```
...
Validating shexmlOutputInstitutions/institutions_5.ttl: 49 conformant, 51 non-conformant
Validating shexmlOutputInstitutions/institutions_6.ttl: 92 conformant, 8 non-conformant
Validating shexmlOutputInstitutions/institutions_7.ttl: 95 conformant, 5 non-conformant
Validating shexmlOutputInstitutions/institutions_8.ttl: 95 conformant, 5 non-conformant
Validating shexmlOutputInstitutions/institutions_9.ttl: 71 conformant, 29 non-conformant
Institutions: 1771 conformant, 524 non-conformant
...
```

## 4.3. Continuous update

Considering that the resulting graph is mainly the output of the batch conversion process, and that the EHRI Portal is continuously receiving new input and ongoing curation [28], there exists a risk of rapidly accumulating a considerable amount of stale data for longer intervals between batch conversion runs. As running the batch conversion takes a considerable amount of resources and time, and can be challenging to operate in a fully unattended manner (mainly due to the possibility of networking and mapping errors, or even infrastructural ones, like power outages), we set out to create a parallel strategy for continuously keeping the data sources synchronised.

Updating materialised data produced by mapping rules is the subject of active research, for instance, IncRML [29] was recently proposed as a way to detect changes on materialised data and handling updates in a surgical manner without rematerialising the whole graph. However, while making use of this approach or even implementing it in ShExML was a possibility – and it will certainly be considered as future work for the presented architecture – we required a method that was straightforward to

implement and, at the same time, easy to modify, flexible (so as to permitting very fine-grained operations), and very efficient.

Taking all these requisites into account, we built upon a system already implemented in the EHRI Portal which exposes a Server-Sent Events (SSE) feed emitting information about lifecycle events such as the creation, update, or deletion of Portal entities. For processing these events, the shexml-streaming[22] library was developed which encapsulates the ShExML engine under a Reactive API and adds support for streaming sources such as SSE or Websockets, or distributed event streaming platforms like Apache Kafka[23]. This allows us to perform all the operations in a non-blocking, asynchronous, event-based manner as the output will be encapsulated in an `Observable`[24] which is to be executed and consumed by the client rather than the library.

From this point, we considered two different options: the use of an ORM-like system for RDF (e.g., DMAOG [30]), or building an *ah-hoc* solution. While the ORM approach would make it possible to avoid creating a lot of task-specific code and would automatically create the necessary SPARQL Update queries, it would also delete triples that should not be removed due to lacking necessary context from the mapping rules that generated them. As a consequence, we created a small module dubbed the EHRI KG Update Service[25].

This service builds upon the previously introduced shexml-streaming library and leverages the `Observable` output to perform its operations in a monoid-like fashion. For each event, the module performs the following operations in sequential order: 1) parses the event data (received from shexml-streaming), 2) downloads the new data from the EHRI Portal GraphQL API, 3) transforms the data using the ShExML engine, 4) transactionally executes the DELETE and/or INSERT SPARQL queries against the SPARQL Update endpoint based on a previously configured SPARQL template (see Listing 8 for an example), and 5) computes the difference between the entity's data before and after the update and outputs a report.

Listing 8: Template used by the update service for the deletion of archival descriptions.

```
PREFIX ehri_units: <http://lod.ehri-project-test.eu/units/>
PREFIX rico: <https://www.ica.org/standards/RiC/ontology#>
PREFIX ehri: <http://lod.ehri-project-test.eu/ontology#>

DELETE {
  ?s ?p ?o .
} WHERE {
  {
     # All the main triples where the unit is the subject except those that are generated as part of
         other set of mapping rules
     ?s ?p ?o .
     FILTER(?s = ehri_units:<$entityId>
        && ?p != rico:hasOrHadSubject
        && ?p != ehri:isCopyOf
        && ?p != ehri:hasCopy
        && ?p != rico:hasCreator)
  }
  UNION
  {
     # Connected instances that can be completely removed as they are only bound to the unit
     ?s ?p ?o ;
        a ?connectedClasses ;
        ?inverseProperty ?unit .
      VALUES ?connectedClasses { rico:Title rico:Activity rico:Identifier }
      FILTER(?unit = ehri_units:<$entityId>)
  }
  UNION
  {
     # Other triples that need to be cleaned up on a per-case basis
     ?s ?p ?o .
     VALUES ?p { rico:isOrWasHolderOf rico:isOrWasTitleOf rico:isBeginningDateOf rico:isEndDateOf rico:
         includesOrIncluded ehri:isOrWasScriptOf rico:isOrWasIdentifierOf }
```

[22]https://github.com/herminiogg/shexml-streaming
[23]https://kafka.apache.org/
[24]https://reactivex.io/documentation/observable.html
[25]https://github.com/EHRI/ehri-kg-update-service

```
        FILTER(?o = ehri_units:<$entityId>)
    }
}
```

This module is highly configurable and even allows for the hot swapping of mapping rules, SPARQL templates or GraphQL queries, making it very amenable to an iterative evolution of the conversion process. However, as detailed in Section 6, this aspect of the architecture is still under active development and we will seek to improve it further and test different approaches.

## 5. Discussion

The architecture presented here is currently being used to populate the EHRI-KG and, while not totally finished, it already contains the large majority of data planned for its first iteration. Presently, the data coming from the EHRI Portal amounts to around 1.2 GB in JSON files. Once converted to RDF the final Turtle file contains almost 7 million triples containing almost 1 million unique entities and 108 different relations. This is, of course, our baseline, and the architecture is implemented, as introduced earlier, in a way that allows for a steady growth of the EHRI Portal and, as a consequence, of the EHRI-KG. In the rest of this section we elaborate on some of the key decisions taken in order to inspire future developments within the community.

The most prominent aspect of this architecture is the orchestration of different sets of mapping rules, a domain which declarative mapping languages and engines themselves have mostly side-stepped, delegating this to other parts of the KGC pipeline [31]. Looking at the Extract Transform Load (ETL) paradigm from an architectural point of view, declarative mapping languages have always fallen under the Transform step, without going further in terms of Extract and Load. While there are some initiatives that try to go a bit further, like the RML-IO module[26] for the new RML specification proposal [32] (which not only defines the `rml:LogicalSource` but also the `rml:LogicalTarget`, hitherto widely relegated to the implementing engine) and the possibility in SPARQL-Generate [33] to call REST APIs in an iterative manner using different parameters, there is a still a common lack of advanced mechanisms that allow performing more complex queries against commonly used endpoints. For example, being able to perform POST requests (in addition to the by-default GET method used by most of the existing mapping languages), where the user can define a custom payload (in this example the GraphQL query) and navigate automatically through a set of paginated results, would reduce a lot of orchestration boilerplate, as well as helping to standardise approaches and make them more reproducible. The recently proposed HTTP Request Access and Dynamic Targets RML extensions [34] seek to implement some of this functionality (e.g., the possibility to perform authenticated POST requests), though it does not seem that they are at the moment able to provide a comprehensive solution for the aforementioned scenario.

Similarly, the possibility to apply a set of mapping rules over a predetermined set of files or endpoints will reduce the efforts needed for merging outputs – not because employing a syntactic merge approach is too cumbersome but as a usability affordance for final users – and could further enable the transformation to be parallelised per source. In ShExML, wildcards[27] grant the possibility to iterate over a set of files, albeit in a fairly inflexible manner, and parallelisation has been recently introduced in the ShExML engine. As disclosed in the documentation, howerver, not all mapping rules benefit equally from the use of parallelisation techniques.[28] Potentially, parallelising on the level of the file will supersede these limitations as we are doing within the orchestration scripts in the presented architecture. Moreover, in order to optimise CPU and RAM usage, the introduction of mini-batches could allow users to pre-define their size, maximise the benefits of a parallel approach and avoid computing bottlenecks, similar to how mini-batches are used in deep learning for gradient descent. Many programming languages APIs already perform this in a heuristic manner like, for example Scala's parallel collections[29] (used by the ShExML

---

[26]https://kg-construct.github.io/rml-io/spec/docs/
[27]https://shexml.herminiogarcia.com/spec/#wilcards
[28]https://github.com/herminiogg/ShExML?tab=readme-ov-file#parallelisation
[29]https://docs.scala-lang.org/overviews/parallel-collections/overview.html

engine) or the Parallel Language-Integrated Query (PLINQ) in C#[30], but offering additional configuration options will help users to optimise their data conversions even further. Parallelisation strategies have been explored in the past by the community, like for example, RMLStreamer [35] which leverages Apache Flink to compute input files or streams in a parallel and streaming manner or Morph-KGC [36] which leverages the concept of mapping partitions to generate disjoints subsets of the target KG, however, empowering users with the possibility to provide fine-grained configurations and decide which aspects are parallelised will help tools to cover different use cases in a more efficient manner.

When dealing with large mapping rules files with a lot of conditions, functions and repeated patterns, the ability to modularise the mapping rules is of great importance. Even in the case of ShExML, whose syntax is fairly succinct in comparison with other mapping languages, the inclusion of the **IMPORT** clause as a way to reuse mapping rules and organise the different aspects of the mapping file, enabled us to have a much more coherent, organised, and maintainable codebase for our conversion architecture, making the process more sustainable and future proof.

Finally, a key step when publishing a KG is to ensure that the generated data is valid. As introduced earlier, SHACL and ShEx can be leveraged for this purpose, or even automatically produced from inputs of one of the processes within the KGC pipeline [31]. Using scripts, or other external tools, as introduced in our architecture, is a viable option, yet requires the user to include this step in their workflow and to analyse the results separately. We contend, therefore, that the incorporation of such feature within the mapping languages and/or engines would ease the burden on users to validate data and make the process more approachable and trustworthy. This could be articulated through the declaration of an external file containing the shapes to which the results of a certain set of mapping rules should conform. Thereby, at materialisation time, the user could already glimpse the validation results and adjust the mapping rules accordingly, in a more interactive and iterative manner.

## 6. Future Work

As it stands, the architecture presented here is able to deliver a comprehensive KG as a semantically-enabled replica of the EHRI Portal, and already fulfils the requirements we set out to deliver. Hence, we do not expect big changes on the overall architecture and our efforts will be mainly dedicated to improving and streamlining some specific aspects to ensure its future sustainability and scalability.

Based on the discussion established in Section 5, our main line of work will be dedicated to the optimisation of the batch conversion, to ensure that it fully utilises all the available resources and reduces, as a consequence, the materialisation time. At present, the Python-based orchestration of the ShExML engine results in repeatedly loading both the JVM itself and the mapping rules (along with any external Scala-based functions), and this adds a non-negligible overhead to the overall process. As a possible solution we will investigate the idea, previously introduced, of using mini-batches for the transformation process which, in combination with the parallelisation over files, could provide some performance improvements.

Similarly, rather than waiting for a GraphQL query to fetch all pages before starting conversion, this task could potentially be triggered immediately after each page download to increase throughput, at the expense, however, of introducing new side effects and potential errors. Therefore, if we were to couple this process, we would investigate declarative ways to traverse endpoints (like REST and GraphQL APIs) which could be integrated in or linked to ShExML, reducing the required orchestration code and the overall complexity, while relying on the inner well-tested coordination mechanisms of the engine as well as enabling the efficient parallelisation discussed above.

In contrast, as the newest layer of this architecture, the update service could evolve rather more, not necessarily because it does not cover the required functionality, but to make it more reliable and easier to maintain. On the reliability side, the main line of work would be the deprecation of the SSE API in favour of a distributed event streaming platform, such as Apache Kafka, that would enable a more distributed and fault-tolerant set-up, a persisted log-based storage (which could additionally serve

---

[30]https://learn.microsoft.com/en-us/dotnet/standard/parallel-programming/introduction-to-plinq

as a source of truth), and the ability to replay events. Moreover, it would allow us to lean more on the new shexml-streaming library and reshape the update layer architecture as a set of data pipelines within Apache Kafka's topics system. As for the maintainability, it once again hinges on the capability to represent this process in a more declarative manner without incurring in additional performance costs. Additionally, the generalisation of the update service will be studied as a means to offer a generic update framework over streaming sources.

Finally, although not germane to the functioning of the conversion architecture, the inclusion of an optional validation step immediately after materialisation will be considered for inclusion in the ShExML engine. This should provide a more interactive framework for developing declarative mapping rules and assist in the earlier detection of issues, especially in combination with the ability to scaffold validation shapes from data mapping rules.

## 7. Conclusions

The architecture presented here has allowed EHRI to provide the data of the EHRI Portal as a KG, making it more interoperable, linkable, aligned with new archival standards such as RiC-O, as well as improving, as a Research Infrastructure, its FAIR compliance. Given the constraints around rebuilding the existing technical stack, the current implementation allows us to maintain an up-to-date replica of the EHRI Portal which also enables new features like the ability to perform arbitrary queries or even executing Federated SPARQL queries throughout linked KGs.

One of the main conclusions of this paper is that even though a great deal of work can be handled by declarative mapping technologies there are still many operations not supported by them, and for which *ad-hoc* solutions are required. However, as discussed above, we see a lot of room for improvement where new features, arising from the necessity of real-world use cases, can enrich the development of these tools and inform new research lines within the field. Until these advancements make their way into practical tools the architecture designed here, and the techniques used to overcome some of the existing limitations and challenges, can inspire other practitioners facing similar scenarios.

By highlighting the deficiencies of existing declarative data mapping techniques we are by no means undermining their value. On the contrary, if this work was at all possible – or achievable in a somewhat short timeframe – it is mainly due to the productivity advantage that these tools provide, making the process more amenable, maintainable and, especially, much more efficient. Therefore, this paper describes our endeavours in bringing a production-ready architecture for the EHRI-KG and builds upon them to signal future advancements in the field which could make these kinds of ventures more feasible and approachable, thereby increasing the productivity of users and reducing project delivery time.

## Supplemental material availability

The source code for the described batch conversion as well as its associated resources are openly available on GitHub (https://github.com/EHRI/ehri-kg-mapping) and under the following permanent DOI: https://doi.org/10.5281/zenodo.17359136. The source code for the update service is available on GitHub: https://github.com/EHRI/ehri-kg-update-service.

## Funding

This work has been carried out in the context of the OSCARS project, which has received funding from the European Commission's Horizon Europe Research and Innovation programme under grant agreement No. 101129751.

## Declaration on Generative AI

During the preparation of this work, the authors used ChatGPT and Gemini in order to: Grammar and spelling check, Paraphrase and reword, and Improve writing style. After using these tools, the authors reviewed and edited the content as needed and take full responsibility for the publication's content.

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
