# OpenReview forum: "Implementing the EHRI-KG: From declarative mapping rules to a three-layered KG construction architecture"
_eswc-conferences.org/ESWC/2026/Workshop/KGCW — KGCW 2026_

### Official Review · ~Els_de_Vleeschauwer1 · 2026-04-02
**Valuable insight in a real-world Knowledge Graph Construction pipeline**

**Rating:** 8
**Confidence:** 4

**Review:**

## Summary
The paper presents a practical architecture for constructing and maintaining the EHRI Knowledge Graph. It integrates batch conversion, validation, and continuous updates, and introduces several useful extensions to ShExML. The work is grounded in real operational constraints and offers valuable insights for the setup of similar knowledge graph construction workflows, while also proposing language‑ and engine‑level improvements to declarative mapping technologies. The topic is highly relevant for the Knowledge Graph Construction Workshop.
## Strengths
- Strong real‑world relevance
- Clear description of the used three layer architecture
- Identification of mapping challenges for declarative mapping languages
- Specification and implementation of solutions in ShExML that address those mapping challenges
- Transparent discussion of the limitations of the current setup and future improvement plans
- Inclusion of the source code
## Weaknesses/ improvement areas
- Lack of quantitative evaluation: The paper does not include empirical measurements of the impact of introduced optimizations or architectural choices. What is the performance benefit of the syntactic file‑level merge compared to re‑serialization? How much materialization time is gained by executing mapping rules in parallel via the Python script instead of running them sequentially, or by doing an update operation instead of a new batch conversion? What is the current end‑to‑end materialization time of the batch conversion, i.e. the baseline for the planned optimizations?
- Fragmented presentation of ShExML extensions: The solutions introduced in ShExML to address the identified mapping challenges—imports, matchers, external functions, and built‑in functions—are spread across Sections 4.1.2 and 4.1.5, making it difficult to gain a complete overview. The title of Section 4.1.5 also risks confusion, since the *future* mapping challenges seem already addressed in ShExML. I recommend consolidating the description of the ShExML extensions in Section 4.1.2—aligning with the logical flow of the batch conversion process, where mapping precedes merging—and using Section 5 to discuss their broader implications for the evolution of declarative mapping languages.
- Incomplete overview of missing features:  Section 4.1.5 mentions that *some* of the missing features in mapping languages, and in ShExML specifically, are highlighted. It is unclear why only *some* are highlighted. Given their relevance for the Knowledge Graph Construction research, I would expect that *all* identified missing features are explicitly enumerated, or at least collected in a referenceable external document.
## Questions
- p. 6: Since the ShExML specification is not versioned, it is unclear what constitutes the “new dimension” of matchers. What were the previous limitations, and what new use cases are now supported?
- p. 8. How does the built‑in function index() relate to RML Logical View indexes (https://kg-construct.github.io/rml-lv/spec/docs/#dfn-index-key)?
## Minor remarks
- Abbreviations without prior introductions, e.g. RLM, ORM
- Typos:
  - p.2: for the their adoption > for their adoption
  - p.7: conformat > conformant
  - p.9: ah-hoc solution > ad-hoc solution
  - p.10: howerver > however

---

### Official Review · ~Sven_Lieber1 · 2026-04-03
**Clearly written paper with interesting use case and qualitative discussion that leaves some open questions**

**Rating:** 9
**Confidence:** 4

**Review:**

## Summary

This is a clearly written paper that introduces a Knowledge Graph construction pipeline with potential for reuse.

The problem statement and use case is sufficiently complex and clear and the work contains many references to (recent) related work which already promises a valuable contribution to the workshop
as it outlines a clear problem related to the state of the art.

Creating a semantic representation of data from an existing platform is as much as important as it is challenging with existing tooling.
From this point of view, the contribution described in the paper is highly relevant.

## Comments
The contribution of the paper is a technical architecture and its implementation.
Yet instead of a more technical evaluation such as a comparison with state of the art tools e.g. via a performance evaluation,
the paper includes a more qualitative evaluation of the overall architecture in the form of a discussion.

Therefore I also focus on the overall architecture and its potential for reuse:


- Among others, the authors cite related work saying that a "comprehensive and extensive methodology is necessary".
Yet the paper does not discuss any methodology in more detail that may have emerged in their use case (besides explanation of the architecture components)
According to me, an important methodological aspect is how to deal with the validation results, how are corrections propagated back to the source?
What kind of issues are actually detectable?

- The authors also cite related work that identifies the need for simple tooling.
Reusable Python mapping files are indeed provided in the repository, they follow clear interfaces which is great!
Yet there is still a lot of boilerplate code in each mapping file that forms a hurdle for less technical users (and maybe even technical users unfamiliar with the used technologies).

- The authors mention that they employ both SHACL and ShEx, but no explanation is given why also SHACL is used.
A sentence or two about why it is necessary would improve the paper.

- Bringing up SHACL, I'm also wondering how the authors deal with subtle changes in the approach behind SHACL and ShEx when interpreting validation results, i.e.
ShEx being a schema language and SHACL a constraint language (see section 7.3 in book.validatingrdf.com/bookHtml013.html )

- From a technical point of view, I'm wondering why the authors create a single huge Turtle file, especially if you also have the option of a streaming solution.
The authors mention that the execution is a bit time intensive, how often is it executed?
Only generating relevant sub parts of the data into e.g. different named graphs or different Turtle files might ease data management tasks. This is also interesting for a possible methodology.
Furthermore, how difficult is the step to "load" the data into a triple store, considering the large size?
Depending on the triple store indexes might have to be built.

- What about error handling? How easy/hard is it to debug?

- Offering a SPARQL endpoint is a good start, yet according to my experiences, using SPARQL is a very high hurdle for non-technical audience.
From a technical perspective, there is also research pointing out the problems of SPARQL endpoints (which are mitigated by the authors with the help of timeouts if I understand correctly).
Is the architecture easily expendible to also include other query interfaces such as LDF?

- What are the limitations of the presented validation approach, especially considering the streaming-based updater?
I'm assuming that not all validation rules are applicable if only a subset of the data is available.


- Related work could also cover workflow systems as eventually Knowledge Graph Construction can be considered a workflow.
For example, the workflow system https://github.com/pegasus-isi/pegasus, active development going back until 2002
Many performance issues might be already addressed when reusing such an existing stack


## Additional suggestions

The authors mentioned that their streaming-based solution works on events triggered by the existing platform.
This indeed is a good start for event-based mapping.
I would like to bring the following to the attention of the authors as it might be relevant:
Wikidata, one of the largest existing knowlege bases, actually stores data internally in a relational database.
Yet the famous Wikidata Query Service that provides a SPARQL endpoint to an RDF representation of Wikidata content,
works on top of a Blazegraph triple store which is also updated continuously.
Their updater-service engine imlements an "event time semantic to re-order the events out of multiple kafka topics"
and "state management consistent with the output of the stream
https://wikitech.wikimedia.org/wiki/Wikidata_Query_Service/Streaming_Updater,
it actually works with RDF diffs, whereas it seems that the authors solution works with Update queries.
It is thus different, but I still would consider it related work.
I also wonder what the tradeoffs are between one and the other approach, this might be interesting future work.


In my experience, a large challenge also includes existing identifiers. Often, catalogues or portals do not provide real persistent identifiers (PIDs),
meaning that if someone deletes a record or merges two records some IDs will disappear and result in 404 errors.
This actually harms FAIR data. Will your KG mirror such mistakes?
How is the proposed architecture dealing with PID-related challenges?
Are old versions kept, will there be some provenance traces? If not why?


For exploration of the RDF data by non-technical users I can recommend setting up a SAMPO-UI platform on top of the RDF data.
(Originally from Finland https://github.com/SemanticComputing/sampo-ui, with recent improvements by Ghent-CDH that eventually will flow back to the main repo https://github.com/GhentCDH/sampo-ui)


## Minor issues
- Section 4 mentions that the architecture is shown in Figure 4, but it is Figure 1.
- Whereas figure 1 provides a complete overview it was not straightforward to comprehend. It took me a while to realize that I have to read it from left to right.
Considering that the text mentions a "three-layered architecture) I tried to read it top-down.
- "for each of these entities, a Graph-QL query template is provided" Do you mean for each of these entity types? I assume you don't have thousands of templates
- what do you mean by "all pages of an entity", what is a page and what specifically is an entity? I assume one "thing", like an institution is one entity and it has one related page. You may have to improve the used terminology

---

### Official Review · ~Ernesto_Jimenez-Ruiz1 · 2026-04-05
**Good effort relevant to the workshop**

**Rating:** 8
**Confidence:** 4

**Review:**

Very relevant KGC in-use effort for the workshop. The paper describes the implementation of a three-layer pipeline to replicate the EHRI portal, currently relying on a Neo4j graph database, with an EHRI-KG using semantic web standards. All the generated materials seem to be public.

Some questions/pointers/suggestions for the authors that may be good to clarify/include in the camera-ready version:
- Will the EHRI-KG supersede the current EHRI? How sustainable is the current architecture in the long term? e.g., if there are major changes in the current EHRI
- Which triple store is used in the new architecture?
- Which ShExML engine is being used? Is there a reference? Or is it the same as footnote 13?
- It would be good to include some example triples and axioms of the ontology to get an idea of the type of data and how it is represented.
- Does the EHRI-KG improve the current EHRI architecture? I miss some quantitative results.

Related/relevant literature:
- Efforts to build and apply a KG in Ecotoxicology:
    - Prediction of adverse biological effects of chemicals using knowledge graph embeddings. Semantic Web 13(3): 299-338 (2022)
- To interpret integrity constraints from OWL ontology axioms:
    - Capturing Industrial Information Models with Ontologies and Constraints. ISWC (2) 2016: 325-343

---

### Decision · Program_Chairs · 2026-04-09

**Decision:**

Accept

**Comment:**

This paper has been selected for presentation at the KGC workshop. We strongly encourage the authors to consider the reviews whilst revising the paper. Camera-ready instructions will soon follow.